# Sustained Endurance Training Leads to Metabolomic Adaptation

**DOI:** 10.3390/metabo12070658

**Published:** 2022-07-16

**Authors:** Astrid Weiss, Katharina Alack, Stephan Klatt, Sven Zukunft, Ralph Schermuly, Torsten Frech, Frank-Christoph Mooren, Karsten Krüger

**Affiliations:** 1German Center for Lung Research (DZL), Cardio-Pulmonary Institute (CPI), Justus-Liebig-University, 35390 Giessen, Germany; astrid.weiss@innere.med.uni-giessen.de (A.W.); ralph.schermuly@innere.med.uni-giessen.de (R.S.); 2Center for Translational and Clinical Research Aachen, RWTH Aachen, 52074 Aachen, Germany; kalack@ukaachen.de; 3Institute for Vascular Signalling, Centre for Molecular Medicine, Goethe University, 60323 Frankfurt am Main, Germany; klatt@vrc.uni-frankfurt.de (S.K.); zukunft@vrc.uni-frankfurt.de (S.Z.); 4Department of Exercise Physiology and Sports Therapy, Institute of Sports Sciences, Justus-Liebig-University, 35390 Giessen, Germany; torsten.frech@sport.uni-giessen.de; 5Faculty of Health/School of Medicine, Witten/Herdecke University, 58448 Witten, Germany; frank.mooren@uni-wh.de

**Keywords:** glycerophosphocholines, cardiopulmonary fitness, athletes, bile acids

## Abstract

Endurance training induces several adaptations in substrate metabolism, especially in relation to glycogen conservation. The study aimed to investigate differences in the metabolism of lipids, lipid-like substances, and amino acids between highly trained and untrained subjects using targeted metabolomics. Depending on their maximum relative oxygen uptake (VO_2max_), subjects were categorized as either endurance-trained (ET) or untrained (UT). Resting blood was taken and plasma isolated. It was screened for changes of 345 metabolites, including amino acids and biogenic amines, acylcarnitines, glycerophosphocholines (GPCs), sphingolipids, hexoses, bile acids, and polyunsaturated fatty acids (PUFAs) by using liquid chromatography coupled to tandem mass spectrometry. Acylcarnitine (C14:1, down in ET) and five GPCs (lysoPC a C18:2, up in ET; PC aa C42:0, up in ET; PC ae C38:2, up in ET; PC aa C38:5, down in ET; lysoPC a C26:0, down in ET) were differently regulated in ET compared to UT. TCDCA was down-regulated in athletes, while for three ratios of bile acids CA/CDCA, CA/(GCA+TCA), and DCA/(GDCA+TDCA) an up-regulation was found. TXB2 and 5,6-EET were down-regulated in the ET group and 18S-HEPE, a PUFA, showed higher levels in 18S-HEPE in endurance-trained subjects. For PC ae C38:2, TCDCA, and the ratio of cholic acid to chenodeoxycholic acid, an association with VO_2max_ was found. Numerous phospholipids, acylcarnitines, glycerophosphocholines, bile acids, and PUFAs are present in varying concentrations at rest in ET. These results might represent an adaptation of lipid metabolism and account for the lowered cardiovascular risk profile of endurance athletes.

## 1. Introduction

Regular training exerts a variety of effects on the metabolism from which the body not only benefits in terms of health but also with an increase in performance. In particular, endurance exercise affects metabolic function, which, depending on intensity and duration, highly challenges the provision of ATP and its re-synthesis by different substrates [1]. When performed regularly, the training induces long-term adaptation processes for optimized and economized energy provision. These adaptations become visible at the level of enzymes, substrate storage, and circulating metabolites [2]. Metabolites are substances that are formed as intermediates or as degradation products of metabolic processes. Particularly during prolonged exertion, lipids and amino acids are increasingly used as substrates for ATP re-synthesis, thereby protecting athletes’ glycogen reserves [3,4].

As an essential adaptation to long-term training, athletes from endurance sports are very well adapted to an effective fat metabolism and the catabolic breakdown of amino acids. The reasons for this adaptation can be found in mechanisms that are initiated to stepwise preserve the glycogen stores by replacing them with other substrates, mainly by the mobilization of free fatty acids, after a progressive depletion of the carbohydrate stores during endurance exercise [4]. Indeed, the ability to conserve glycogen stores through adaptations of lipid metabolism might be one of the performance-determining factors of long endurance athletes [5].

Hence, an optimised production and usage of specific lipids would be expected as an adaptation to regular endurance training. These lipids include phospholipids and sphingolipids for which previous studies have proven that their concentration in blood changes after exercise and training [6]. Regarding preventive and therapeutic approaches to exercise training, the effect of physical activity on these lipids is particularly significant for the positive effect of physical activity on dyslipidaemia and, through their accumulation in the muscle, for preventing insulin resistance [2]. The increased energetic demand during athletic exertion also ensures an increased tricarboxylic cycle (TCA) flux in skeletal muscle, resulting in the increased formation of acetyl-carnitine. Plasma acylcarnitines attract interest because they may represent biomarkers of metabolically acquired conditions, such as diabetes, obesity, insulin resistance, and cardiovascular diseases [7]. Bile acids synthesized by the liver are responsible for the breakdown of ingested fats. Regarding its chemical structure, bile acids are steroid carboxylic acids derived from cholesterol. The primary bile acids are cholic and chenodeoxycholic acids. They are conjugated with glycine or taurine before they are secreted into the bile fluid. There is evidence that endurance exercise acutely decreases the concentration of total bile acids in serum, which is interpreted as a favorable effect of physical exercise for preventing tissue damage [8]. However, the corresponding data is only limited and the physiological backgrounds are speculative.

Metabolomics as a research approach allows a large amount of low-molecular metabolic compounds in a biological system to be analyzed in a high-throughput format. Targeted metabolomics describes the simultaneous identification and quantification of a pre-selected set of metabolites. Therefore, an assay is used that comprises distinct metabolites of one or more metabolic pathways of interest. Due to their potential importance in the adaptation to endurance sports and general health significant amino acids, biogenic amines, acylcarnitines, lysophosphatidylcholines, phosphatidylcholines, sphingolipids, polyunsaturated fatty acids (PUFAs), oxylipins, and bile acids were targeted [9,10].

The aim of the present study was to investigate the metabolic profile of trained endurance athletes compared to un-trained controls with a focus on lipids, bile acids, and amino acids. It was considered that the specific dietary habits of athletes might have a significant influence on the metabolome. Furthermore, associations of individual metabolites with maximum oxygen uptake were examined to point out possible direct relationships with cardiopulmonary fitness.

## 2. Materials and Methods

The present manuscript is based on a previous study of our group investigating the effects of endurance training status on kinase activity and apoptosis sensitivity in lymphocytes [11,12]. The plasma samples were obtained from the same subjects. We refer to this work regarding anthropometric and physiological data, leukocyte cell count, and lymphocyte subpopulation [11].

### 2.1. Ethical Approval

The study was carried out in accordance with the Declaration of Helsinki. All experimental procedures were approved by the local ethics committee of the Justus-Liebig -University Giessen (Germany). Before study participation a written declaration of consent was signed by all subjects.

### 2.2. Study Design

First, only male subjects were checked for their unrestricted participation in sports, and the anthropometric data were collected in a mandatory medical examination. Second, an endurance exercise capacity test was conducted using a continuous incremental exercise protocol on the treadmill as previously explained in detail [11]. Participants were categorized as either endurance-trained (ET; *n* = 12; VO_2max_ =67.15 ± 1.51 mL/kg × min^−1^ [Mean ± SEM]; 28.58 ± 1.98 years [Mean ± SEM]) or un-trained (UT; *n* = 11; VO_2max_ = 40.43 ± 1.24 [Mean ± SEM]; 30.55 ± 2.41 [Mean ± SEM]) based on their maximum relative oxygen uptake (VO_2max_). A detailed presentation of the anthropometric and physiological data and all inclusion and exclusion criteria has already been published by our group [11]. Subjects were included in the endurance-trained group if they achieved a relative VO_2max_ ≥ 59 mL/kg × min^−1^ and categorized as UT if they had a relative VO_2max_ ≤ 45 mL/kg × min^−1^. At least seven days after the exercise testing, a standardized venous blood collection was performed under resting conditions. An overview of the study course and the methodological workflow of the metabolomic phenotyping is included in the Results Section.

### 2.3. Nutritional Status

The subjects recorded the amount of all foods and beverages consumed over a period of seven days. The additional intake of food supplements was not documented. The DGExpert Software of the German Nutrition Society (DGE) was used for the analysis of the average weekly nutrition intake.

### 2.4. Blood Sampling Procedure

The subjects were instructed to renounce vigorous physical activity 24 h and food intake six hours prior to blood sampling. The standardized blood collection procedure from a peripheral vein was always carried out from 8 to 11 a.m. under conditions of physical rest. The ethylene diamine tetra acetic acid (EDTA) plasma was separated from cells by centrifugation at 20 °C for 10 min at 2500× *g*. Next, aliquots of 500 µL plasma were transferred into pre-cooled storage vials for the analysis of endogenous metabolites and bile acids (Biozym, Biozym Scientific GmbH, Oldendorf, Germany). For the study of polyunsaturated fatty acids (PUFAs), plasma was aliquoted in sampling tubes with butylated hydroxytoluene (BHT) (SPI Bio, Bertin Pharma, Montigny-le-Bretonneux, France). The plasma aliquots were frozen immediately and stored at −80 °C until extraction.

### 2.5. Targeted Metabolomics

Plasma metabolites were extracted by using the following three assays: (1) Biocrates AbsoluteIDQ p180 kit (BIOCRATES, Life Science AG, Innsbruck, Austria), (2) AbsoluteIDQ Bile Acids kit (BIOCRATES, as above), and (3) PUFA assay established at the Metabolomics Core Facility at the Goethe University (Frankfurt am Main, Germany). The AbsoluteIDQ p180 kit allows the detection of 188 metabolites, including 42 amino acids and biogenic amines, 40 acylcarnitines, 90 glycerophosphocholines, 15 sphingolipids, and the sum of hexoses, whereas the AbsoluteIDQ Bile Acids kit includes 20 bile acids. Metabolites were extracted according to the manufacturer’s protocols. Further, the provided multiple reaction monitoring (MRM) tables of Biocrates were used for metabolite identification and as previously described [13,14]. The PUFA assay was performed the following way: Plasma was mixed with 1 mL of ethyl acetate, samples were shortly vortexed, centrifuged at 15,000× *g* for 1 min, and the upper layer was transferred into a clean tube. The extraction step was repeated (addition of 1 mL ethyl acetate with 0.1% formic acid) and the two upper layers were combined. The extracts were vortexed and centrifuged again, and the supernatants were dried in a vacuum under a nitrogen stream. Last but not least, samples were reconstituted in 50 µL of AcN:H_2_O (1:1, *v*/*v*) and transferred to MS glass vials ready to be analyzed by the mass spectrometer. Negative ionization ESI-LC MS/MS was performed on an Agilent 1290 Infinity LC system (Agilent, Waldbronn, Germany) coupled to a QTrap 5500 mass spectrometer (Sciex, Darmstadt, Germany). Ion source parameters were as follows: CUR 20 psi, CAD medium, Ion Spray Voltage-4500 V, TEM 525 °C, GS1 65 psi, and GS2 70 psi. In total, 137 oxylipins (all purchased from Cayman Chemical) were included in this targeted MRM screen. Oxylipins were identified with authentic standards and/or via retention time, elution order from the column, and 1–2 transitions. Reversed-phase LC separation was performed by using an Acquity UPLC BEH Shield RP C18 1.7 µm (2.1 × 100 mm) column. Compounds were eluted with a flow rate of 0.5 mL/min and with the following 8 min gradient: 0 min 99.9% A, 0–4.5 min 45% A, 4.5–5 min 1% A, 5–5.8 min 1% A, 5.8–5.9 min 99.9% A, and 5.9–7.9 min 99.9% A. Solvent A consisted of H_2_O:AcN (6:4, *v*/*v*) containing 0.02 % acetic acid, and solvent B consisted of AcN: Isopropanol (1:1, *v*/*v*). The column oven temperature was set to 40 °C, and the autosampler was set to 4 °C. The injection volume was 8 µL. Additionally, eminent metabolite ratios with known biological functions were analyzed. Concentrations for most metabolites are given in µM or in ng/mL, if not mentioned otherwise.

### 2.6. Statistical Analysis

The analysis of anthropometric and physiological data as well as leukocyte cell count was performed by SPSS Version 23 (IBM^®^SPSS Statistics, IBM GmbH, Munich, Germany) and GraphPad Prism 5.01 (GraphPad Software, La Jolla, CA, USA) [11]. Parts of those data are now presented in diagrams (Figure 1), which were generated using a newer software version. As for all other results, too, statistical analyses were performed by using GraphPad Prism version 8.4.3 (686) (GraphPad Software, LA Jolla, CA, USA) or by the MetaboAnalyst platform (version 5.0, (https://www.metaboanalyst.ca/MetaboAnalyst/home.xhtml), which uses an R package (MetaboAnalystR: https://github.com/xia-lab/MetaboAnalystR) for statistical and functional analyses (all accessed on 4 August 2021). Details of the selected statistical methods for each displayed figure or table are described in the accompanying legend.

## 3. Results

### 3.1. Study Design and Descriptive Parameters

Participants were enrolled on voluntary basis and had to undergo a medical examination prior to categorization into both study groups, namely endurance-trained (ET) athletes or un-trained (UT) individuals (Figure 1A). As already described [11], significant changes in several anthropometric measures could be detected. Most importantly and as expected, body weight, body mass index (BMI), and body fat were strongly reduced in the ET group compared to the individuals of the age- and height-matched UT group (Figure 1B and Appendix A). In line with this, endurance exercise testing revealed significant differences in several parameters, reflecting the intensity of exercise, such as weekly training sessions as well as training durations and cardio-pulmonary performance as reflected by the relative VO_2max_ (maximal oxygen uptake) (Figure 1C and Appendix A). Based on those criteria with a strong emphasis on relative VO_2max_, the participants were categorized into ET athletes with a relative VO_2max_ higher than 59 mL/kg × min^−1^ or UT individuals with a relative VO_2max_ less than 45 mL/kg × min^−1^. At least seven days of rest after spiroergometry, blood was collected from all participants at resting conditions. The plasma was separated and stored at −80 °C until further processing and metabolomics analyses. By this approach, the global effect of chronic sustained high intensity endurance exercise on the plasma metabolome should be observed.

### 3.2. Nutritional Status

To exclude any possible interference by nutritional status, all participants had to provide a nutrition diary based on a standardized questionnaire that comprised a period of seven days. In addition, subjects were not allowed to consume any food six hours prior to blood draw. Software assisted calculations allowed the comparison of the energy balance for both study groups, which showed significant differences in sugar alcohols and dietary fibres (*p*-values of *: *p* ≤ 0.05), both of which are slightly enhanced in the ET group (Figure 2A). In more detail, most significantly deregulated nutrients including dietary fibres, copper, fluoride, magnesium, water, and vitamins display an increased uptake for the ET group, except for eicosatrienic acid, the ratio of hexadecenic to palmitoleinic acid, and saturated fatty acids. Latter ones are taken up to a larger extent by the UT group (Figure 2B; *p*-values of *: *p* ≤ 0.05; **: *p* ≤ 0.01; ***: *p* ≤ 0.001). These differences need to be considered during the interpretation of the upcoming results from the metabolomic analyses as the nutritional status can also have, in principle, an impact on the plasma metabolome composition.

### 3.3. Targeted Metabolomics Analyses

Plasma levels of distinct metabolites were quantified by LC-MS/MS and computational analyses. Figures were generated by using the freely accessible MetaboAnalyst software (Version 5.0). Data were generated in three independently conducted assays: In the first assay, amino acids (in total 21), biogenic amines (in total 21), hexoses, acylcarnitines (in total 40), glycerophosphocholines (in total 90), and sphingolipids (in total 15) were extracted from plasma and quantified by using the AbsoluteIDQ p180 kit (Biocrates). The results are summarized in Figure 3 and Figure 4. A two-dimensional PCA scores plot (principal component analysis; Figure 3A) and a PLSDA plot (partial least squares discriminant analysis; Figure 3B) revealed a clear separation of both study groups, between ET and UT individuals. Next, samples were arranged in a heat map (data normalization via median and log_2_), according to their group affiliation with increasing relative VO_2max_, displaying the top 30 deregulated metabolites (or their ratio) (Figure 3C). Clusters of increased metabolites in ET athletes can be detected in the upper left area of the heat map, while metabolites with decreased plasma concentration are localized at the lower left. Plotting the log_2_ fold change (FC; as calculated from the respective group means for each metabolite plasma concentration before data normalization) against the −log_10_ (*p*-value) allows visualization of metabolites that are strongly under-represented in ET (blue labels, 1 to 4) or over-represented in UT (red labels, 5 to 7) together with the significance of the respective t-test analyses (visualization via volcano plot, Figure 4A). Figure 4B summarizes significant concentration differences (in µM) of seven metabolites. Further, a summary of the corresponding analysis, including the identifier in the human metabolome database and general category of the metabolites, is shown in Table 1. The lipid names indicate the respective linkage by an ester (i.e., a for acyl), two acyl (i.e., aa for diacyl), or by a combination of an ester and an ether (i.e., ae for acyl and alkyl). In detail, significant changes were detected for one sphingomyelin (SM C22:3, down in ET, #2), one acylcarnitine (C14:1, down in ET, #3), and five glycerophosphocholines (lysoPC a C18:2, up in ET, #5; PC aa C42:0, up in ET, #6; PC ae C38:2, up in ET, #7; PC aa C38:5, down in ET, #1; lysoPC a C26:0, down in ET, #4) in endurance-trained athletes compared to un-trained individuals. Nevertheless, these findings clearly demonstrate that frequent high intensity endurance exercise can also influence the plasma metabolome when the time point of sampling is independent of any type of acute physical activity. In the second assay, bile acids (20 x) were extracted from plasma and quantified by using the AbsoluteIDQ Bile Acids kit (Biocrates). The results are summarized in Figure 5 and Figure 6. Although both cohorts did not perfectly separate in the 2D scores plot, the 3D PLSDA plot shows that ET and UT can indeed be distinguished from each other (Figure 5A,B). However, the overall difference in plasma bile acid composition is too small to give rise to any cluster formation in the top 30 heat map (Figure 5C). A final data comparison via volcano plot (same thresholds as before) revealed four significantly altered metabolites (or ratios) in ET compared to UT (Figure 6A, highlighted). In detail, TCDCA (#1) was down-regulated in ET athletes while three ratios (CA/CDCA, #3; CA/(GCA+TCA), #2; DCA/(GDCA+TDCA), #4) were up-regulated (Figure 6B, box plots, in µM). Table 2 summarizes the corresponding statistical description and HMDB IDs. The highest fold changes were detected in the ratios of DCA to GDCA+TDCA, with more than a 4-fold increase in ET compared to UT. Furthermore, the ratios of CA to CDCA and CA to GCA+TCA also demonstrated a fold change of 2.65 and 1.92, respectively. In line with this, the change of TCDCA by 0.38-fold is comparable to the alterations observed in the case of the deregulated metabolites from the first measurement. In the third assay, eicosanoids, fatty acids, and PUFAs were extracted from plasma and quantified by using an established method of the metabolomics facility (see Methods Section for details). The results are summarized in Figure 7 and Figure 8. Again, the 2D scores plot (Figure 7A) as well as the PLSDA plot (Figure 7B) clearly showed separation of both study groups, which also becomes visible in the top 30 heat map (Figure 7C) by the formation of clusters. Metabolites with enhanced plasma levels in ET athletes compared to UT participants are located in the upper left area of the heat map, while candidates with a diminished concentration can be seen in the lower left region. With respect to significance of these alterations, only three metabolites are of further interest that are highlighted in the volcano plot (Figure 8A). TXB2 (#1) as well as 5,6-EET (#2) are both eicosanoids and downregulated in the ET group versus UT and 18S-HEPE (#3), a PUFA, is upregulated (Figure 8B). All descriptive statistical parameters are shown in Table 3. Most interestingly, TXB2 is reduced 0.11-fold in ET compared to UT, with a high significance (*p* = 0.001), while 5,6-EET levels are 0.66-fold lower. With a fold change of 1.72, 18S-HEPE is strongly upregulated in the group of ET athletes. In line with the above-mentioned analyses, it can be concluded that regular intense endurance training and accompanied cardio-pulmonary fitness leads to adaptations of the plasma metabolome affecting physiological processes involving fatty acid oxidation, lipid metabolism, glycemic control, energy expenditure, and immune modulation as well as the regulation of cardio-pulmonary functions. This confirms and further supports the concept of the metabolic benefits of exercise.

### 3.4. Correlation between Plasma Metabolites and Endurance Performance

In this study, the enrollment of participants into the group of ET athletes is based on their relative VO_2max_, which must be greater than 59 mL/kg × min^−1^, while a value of less than 45 mL/kg × min^−1^ was used to categorize persons into the group of UT individuals. This physiological parameter was obtained during spiroergometry, as previously described [11]. It is used as an indicator for cardio-pulmonary fitness. In our aforementioned analyses, the focus was solely on the overall difference in plasma metabolite concentrations between those two groups independently from the individual relative VO_2max_, i.e., the personal endurance performance. Following that, correlation analyses between all metabolites and the individual relative VO_2max_ values regardless of the participants´ group affiliation were conducted. In detail, metabolites from all three above-mentioned assays have been investigated for their direct association with individual relative VO_2max_ values. For this purpose, raw data from all metabolite concentration underwent a correlation analysis with all 23 individuals relative VO_2max_ values. Assuming a Gaussian distribution, a statistical two-tailed t-test was applied and the Pearson correlation coefficient was computed for each metabolite data point versus the respective participant´s individual relative VO_2max_ value. Only significant correlation results are displayed (Table 4), including HMDB ID, category, and statistical parameters. The amino acids aspartate and glutamate showed a negative correlation with relative VO_2max_, while the ratio of citrulline/ornithine demonstrated a positive correlation. The two biogenic amines, α-AAA and taurine, are negatively associated with relative VO_2max_, and this similarly holds true for two sphingomyelins (SM C18:0, SM C18:1) as well as for one acylcarnitine (C16). Within the group of glycerophosphocholines, a mixed pattern of correlation has been obtained. For the phosphatidylcholine PC aa C38:4, an inverse correlation with relative VO_2max_ was shown. For PC ae C38:2, PC ae C42:2, lysoPC a C18:1, and for the ratio of lysoPC_total_ to PC_total_, their plasma concentrations positively correlated with the relative VO_2max_ values. Two bile acids, namely GCDCA and TCDCA, both showed a clear negative correlation, while the ratio of CA to CDCA as well as both PUFAs (LXA4 and 18-HEPE) are positive correlators with relative VO_2max_. These analyses clearly demonstrate that there are changes of certain plasma metabolites even at resting conditions that are directly linked to the endurance performance regardless of any other physiological parameter and without a pre-classification into an endurance-trained athlete or an un-trained participant. This means that there might be metabolites that continuously circulate in the plasma, which could be used as easily accessible indicators of cardio-pulmonary fitness as acquired by frequent high intense endurance exercise sessions.

### 3.5. Identification of Biomarkers for Cardio-Pulmonary Fitness

Targeted metabolomic analyses revealed a substantial number of different metabolites that are deregulated in the group of ET athletes compared to UT participants or that are direct correlators with the marker of endurance performance, i.e., relative VO_2max_, both at resting conditions. Finally, it would be of interest to identify candidates that fulfill both criteria by demonstrating a significant change in its plasma concentration between ET versus UT on the one hand and which are directly associated with cardio-pulmonary fitness. By comparing the lists of deregulated metabolites between the two participant groups (ET versus UT) (Table 1, Table 2 and Table 3) with the list of direct relative VO_2max_ correlators (Table 4), two metabolites and one ratio of bile acids were found to meet both criteria. The phosphatidylcholine PC ae C38:2, the bile acid taurochenodeoxycholic acid (TCDCA), and the ratio of cholic acid to chenodeoxycholic acid (CA/CDCA) are differentially regulated in ET athletes compared to UT individuals and their concentrations are closely linked to the respective relative VO_2max_ value. Raw data plasma levels for each of these metabolites (Figure 9A,B) as well as for the ratio of the given bile acids (Figure 9C) allow an easy estimation of the mathematical relationship, as indicated by the linear regression calculation. In addition, the correlation matrix provides the Pearson r correlation coefficients and significance levels for the correlation amongst the three metabolites (Figure 9D). Interestingly, there exists a significant relationship in each case of correlation only with the exception for the comparison of TCDCA and CA/CDCA. Nevertheless, the fact that a set of metabolites with different biological origin and function directly correlate with the same physiological parameter, i.e., relative VO_2max_, demonstrates that these candidates should be considered as plasma biomarkers of cardio-pulmonary fitness, especially when measured in a combined mode.

## 4. Discussion

The aim of the present study was to investigate the metabolic profile of endurance athletes compared to un-trained individuals based on selected metabolites from lipid and amino acid metabolism by means of a targeted metabolomics analysis. Despite a relatively low number of subjects, we could demonstrate that there are metabolites significantly reduced in endurance athletes or increased, respectively, and/or even associated with VO_2max_ across both groups. While SM C22:3, C14:1, PC aa C38:5, and lysoPC a C26:0 were found reduced in athletes, lysoPC a C18:2, PC aa C42:0, PC ae C38:2, and 18S-HEPE were increased in serum. Regarding bile acids, TCDCA was down-regulated in athletes, while for three ratios of bile acids namely CA/CDCA, CA/(GCA + TCA), and DCA/(GDCA + TDCA) an up-regulation was found. Within these metabolites, PC ae C38:2, TCDCA, and the ratio of cholic acid to chenodeoxycholic acid (CA/CDCA) showed an association with VO_2max_ across groups, implicating their potential as biomarkers for cardiopulmonary fitness in healthy individuals. This metabolic adaptation is most likely related to the adjustments of energy metabolism in endurance athletes, reflecting the substrate utilization for repetitive prolonged bouts of exercise. In these contexts, one major adaptive factor known as mobilization and metabolization of lipids, lipid-like molecules, and amino acids appears to increase. This may also be associated with positive health outcomes, which is confirmed by the fact that endurance athletes have a rather low cardiovascular risk. For some of the metabolites analyzed here, however, an increase in the resting concentration in the blood could also be an indication of an increased cardiovascular risk; so, we can expect a downregulation in endurance athletes [15].

There are many different molecules of the above-mentioned types in human blood, so it was expected that a specific profile of these metabolites might reflect the difference in training status. Since a key differentiator between the UT and the ET group is cardiopulmonary fitness, we hypothesised that there would also be cross-group associations with fitness. In accordance with these assumptions, group distinctions were first determined within phospholipids, acylcarnitines, and glycerophosphocholines. Sphingomyelins represent one of the major components of cell membranes, and they are synthesized through the transfer of a phosphorylcholine head group from phosphatidylcholine to ceramide. It was previously shown that plasma SMs are associated with cardiovascular and metabolic diseases [16]. In coherence with these data, negative associations with the level of cardiovascular fitness have already been found for some SMs, which correlates with our findings for SM C22:3 [17].

PCs represent a group of phospholipids that have already been associated with cardiorespiratory fitness. However, this was mainly in the elderly or specific patient groups, and it often also served to assess the cardiovascular risk profile [18]. For example, for PC C38:2, an inverse correlation with a waist gaining phenotype in women was shown, which might indicate a relation to the capacity of energy storage [19]. If weight gain is positively associated with this metabolite, the high energy turnover could possibly be explained by a negative regulation of this PC.

LysoPC a C18:2 represents a lysophosphatidylcholine in which the single acyl group contains 18 carbons and 2 double bonds. An increase of this metabolite has already been shown for endurance athletes compared to control subjects and strength athletes [20]. Regarding acute endurance exercise, lysoPC a C18:2 was found to be elevated after a marathon race in subjects with a high VO_2max_ [21]. Therefore, an increase of this metabolite might indicate a condition of challenges in metabolic performance. An association with cardiopulmonary fitness levels is supported by findings from patients with heart failure, where lysoPC a C18:2 was found to be lower compared to healthy controls. A metabolic stimulus is probably missing here, as these patients are usually very inactive [22]. An association of PC aa C42:0 with cardiopulmonary fitness has already been shown for women in the KarMeN Study [23], which strengthens the assumption that it is a cross-gender marker of cardiopulmonary fitness.

Acylcarnitines serve as transport carriers for fatty acids when they are transferred into mitochondria. Here, the metabolization of fatty acids takes place via β-oxidation. Without the transport molecule, i.e., carnitine, the outer mitochondrial membrane would be impassable for the fatty acids. Our data indicate that C14:1 was reduced in the blood of athletes. This is quite surprising as C14:1 is involved in β-oxidation of long-chain fatty acids. A recently published study showed that the C14:1 level significantly increased in plasma in response to acute endurance exercise. This rise was even more pronounced in endurance athletes than in controls or strength athletes [20]. Since we did not study acute exercise but the resting state of endurance athletes, we cannot find a clear association with our data.

Bile acids are involved in intestinal nutrient absorption and biliary secretion of lipids. They also represent signaling molecules and metabolic regulators of hepatic lipid, glucose, and energy homeostasis [24]. There is little data on bile acids and exercise, and some of the results are even contradictory. For TCDCA, no response was found to acute strength training, while some studies report an increase, others report a slight decrease after acute endurance exercise [8,25]. TCDCA represents one of the main effective components of bile acids, which has shown an involvement in the regulation of inflammation via a dynamic interaction with NF-κB. Since any intense acute physical stress is a pro-inflammatory stimulus with an anti-inflammatory counter-regulation, this could explain the downregulation of this bile acid in athletes [26]. The regulation of cAMP is also linked to this bile acid, so that regular energetic stress could also influence the release of TCDCA [27]. CA and CDCA represent primary bile acids, which are synthesized from cholesterol in the liver, and they are converted to secondary bile acids (DCA and TCA) in the intestine [28]. Both bile acids showed a reduction after acute exercise, including after a half-marathon [8,25]. **CA** drops particularly sharply here, which may explain the reduced quotients. This makes sense against the background that increased levels, including serum concentrations, of numerous bile acids increase the risk of diseases of the GI tract, and endurance training could counteract this. We can only speculate about the physiological causes of the reduction in bile acids. The lower serum concentrations of cholesterol could reduce the effective synthesis of bile acids. Especially, the synthesis of non-conjugated bile acids could be affected, while others, such as TCDCA and GDCA, were not reduced in other exercise studies [8]. Another cause may be a difference in the gut microbiome of athletes. Some of the bile acids are regulated by bacteria in the microbiome. Some studies show that athletes have an increased α-diversity in the microbiome and that some bacteria are more abundant in athletes, such as bifidobacterium, bacteroides or veillonella [29]. The study of the athlete microbiome in relation to the blood metabolome is a focus of future research projects.

TXB2 as well as 5,6-EET represent PUFAs that were found to be reduced in athletes. These metabolites are currently being discussed in the context of sports activities as lipid mediators that are part of the exercise-induced immune response. Accordingly, it could be shown for TXB2, for example, that acute sporting activities lead to upregulation and also to increased platelet activation [30]. However, the reduced concentrations in athletes at rest could be the result of pro-resolution functions, following acute inflammation. The decreased levels of TXB2 might also reflect a lower synthesis of its precursor, thromboxane A2, which is involved in platelet activation and aggregation [31]. Accordingly, this could reflect the improved fibrinolytic profile induced by regular training [32]. We further demonstrated that 18-HEPE was found at higher concentrations in the plasma of athletes. 18-HEPE is produced by non-enzymatic oxidation of eicosapentaenoic acid (EPA), and it represents an anti-inflammatory and anti-fibrotic metabolite. The reason 18-HEPE circulates in elevated concentrations in athletes relative to inactive persons cannot be deduced from the current data. However, it has been recently reported that the regression of human coronary artery plaques is associated with a high ratio of 18-HEPE + resolvin E1 to leukotriene B4 [33], which further supports the hypothesis of an anti-inflammatory environment induced by exercise. As with the other lipids and the bile acids, we cannot rule out the possibility that the different dietary habits of the athletes are responsible for some of the differences. Thus, the analysis of food intake revealed that the athletes showed significant deviations in food intake. In particular, lipid fractions showed very differentiated deviations, which could be an influencing factor for both namely increased and reduced metabolites, such as phospholipids, glycerophosphocholines, eicosanoids, or bile acids. Accordingly, in addition to physiological adaptations, this can also be an explanation for the different concentrations [34,35].

In summary, we have shown that numerous phospholipids, acylcarnitines, glycerophosphocholines, bile acids, and eicosanoids are present in varying concentrations at rest in subjects with high cardiovascular fitness. This seems to be mainly due to the regular metabolic stress and the different substrate utilization that takes place during endurance exercise. In addition to reflecting these group differences, some of these correlations with cardiopulmonary fitness as determined by VO_2max_ also show up across both groups. Future studies in larger collectives/cohorts would have to re-examine the extent to which individual metabolites can be used as biomarkers for diagnosing cardiopulmonary fitness and thus for also estimating possible cardiopulmonary or metabolic risks.

## Figures and Tables

**Figure 1 metabolites-12-00658-f001:**
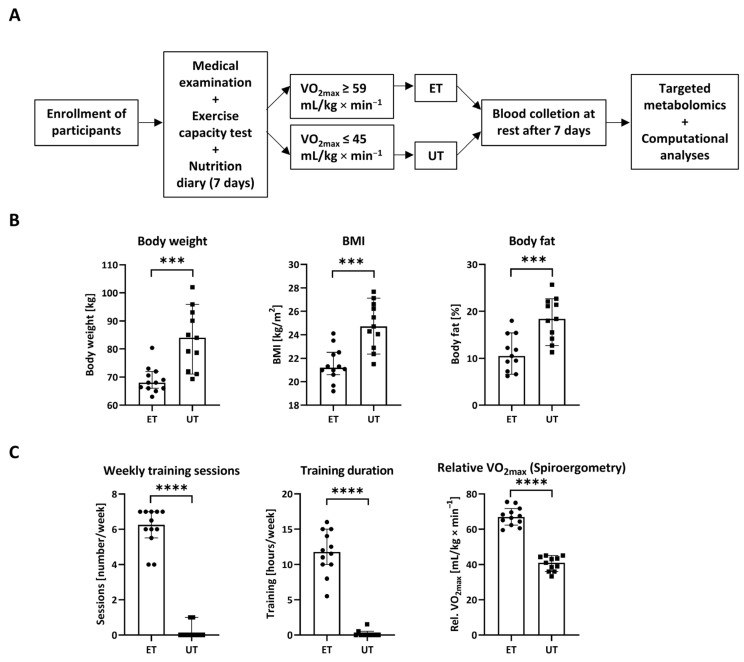
Study setup and principle. (**A**) Male volunteers were enrolled and underwent medical examination and exercise testing prior to categorization into endurance-trained (ET) athletes or un-trained (UT) individuals based on relative VO_2max_. In addition, nutrition diaries using a standardized questionnaire had to be provided over a time period of seven days. Blood was taken from all participants one week after spiroergometric assessment. Once 14 male subjects per group were successfully recruited, samples without outliers due to technical issues were analyzed via targeted metabolomics. Finally, software-assisted analyses were conducted to find out significant metabolic differences between ET athletes and UT participants. (**B**) Physiological parameters, i.e., body weight, body mass index (BMI), and body fat are significantly reduced in ET athletes (*n* = 12) versus UT individuals (*n* = 11). (**C**) Data describing the frequency and the intensity of endurance exercise as well as the main discriminator, i.e., relative VO_2max_, are given. Data have been statistically analyzed by an unpaired, non-parametric, two-tailed *t*-test (Mann–Whitney test). ***: *p* ≤ 0.001; ****: *p* ≤ 0.0001.

**Figure 2 metabolites-12-00658-f002:**
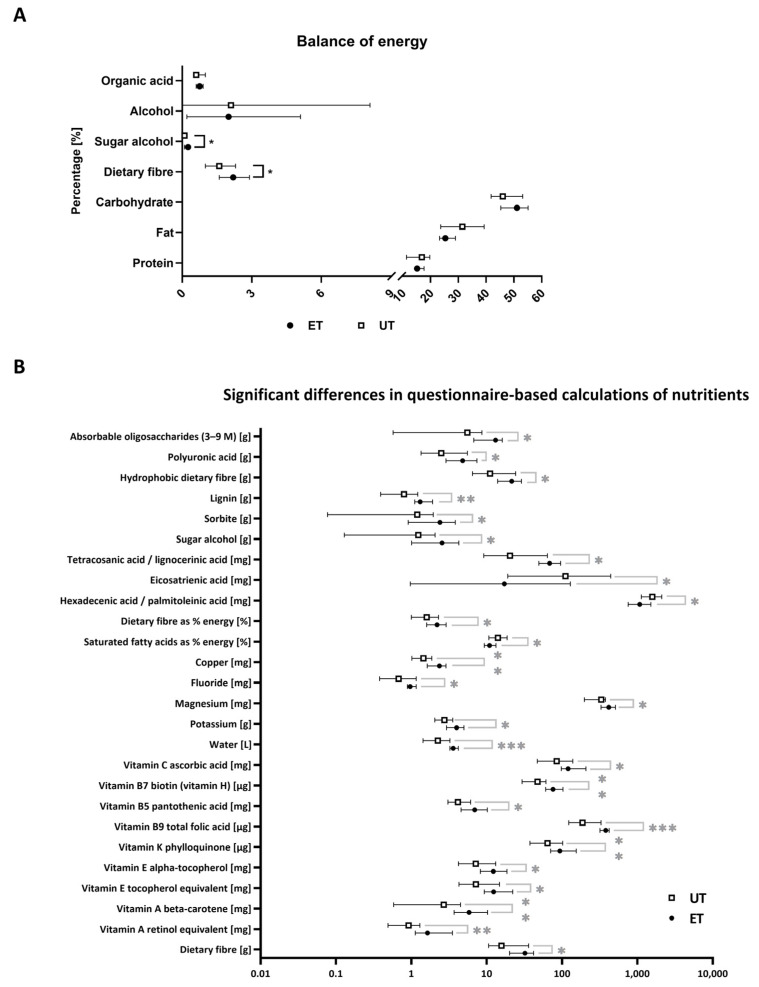
Nutritional status. (**A**) Combined presentation of the questionnaire-based nutritional status with regard to the basic source of energy given in percentages. (**B**) List of nutrients that were taken up in significantly different concentrations as calculated by the subjects’ questionnaires. Data have been statistically analyzed by an unpaired, non-parametric, two-tailed *t*-test (Mann–Whitney test). *: *p* ≤ 0.05; **: *p* ≤ 0.01; ***: *p* ≤ 0.001.

**Figure 3 metabolites-12-00658-f003:**
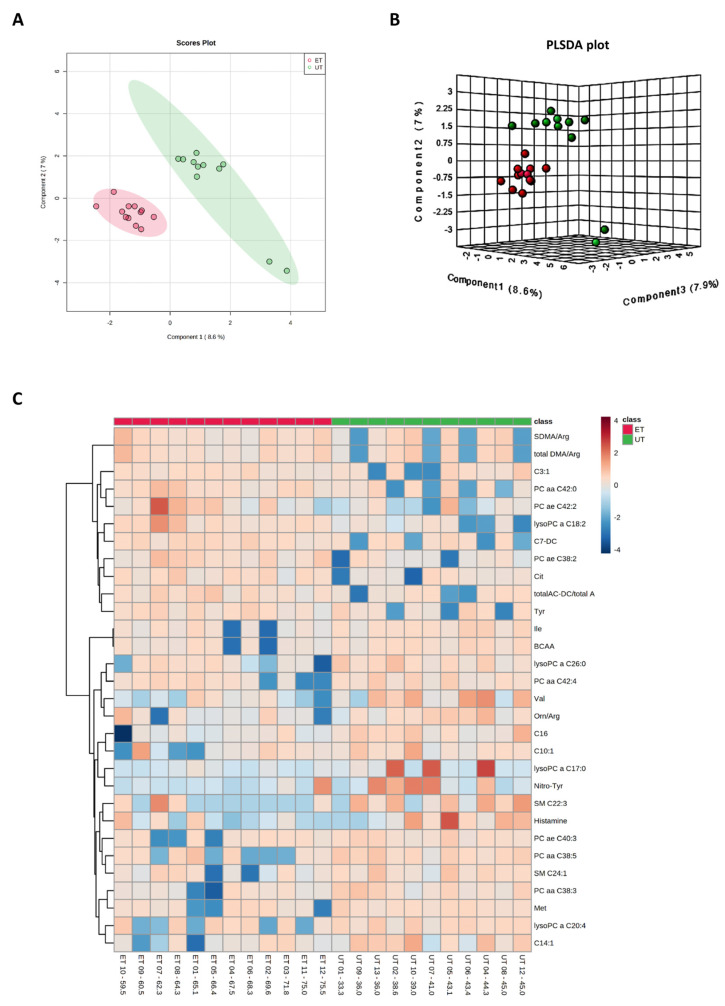
Component and heat map analyses of metabolites detected by the Biocrates Absolute IDQ p180 kit. (**A**) 2D principal component analysis (PCA) scores plot displaying individual samples from both groups, i.e., from ET (pink) and UT (green) subjects. (**B**) 3D multiple component partial least squares discriminant analysis (PLSDA) plot showing individual samples from both groups, i.e., from ET (red) and UT (green) subjects. (**C**) The heat map only shows the top 30 deregulated metabolites or ratios as listed on the vertical dimension. Metabolites with high plasma levels are highlighted in orange to red while under-represented metabolites are colored in light to dark blue. Therefore, raw data, i.e., metabolites´ concentrations, had to be normalized by their median and log_2_-transformed prior horizontal arrangement according to their group affiliation with increasing relative VO_2max_. Samples from ET athletes are located on the left side with red labels, and samples from UT subjects can be identified by their green labels.

**Figure 4 metabolites-12-00658-f004:**
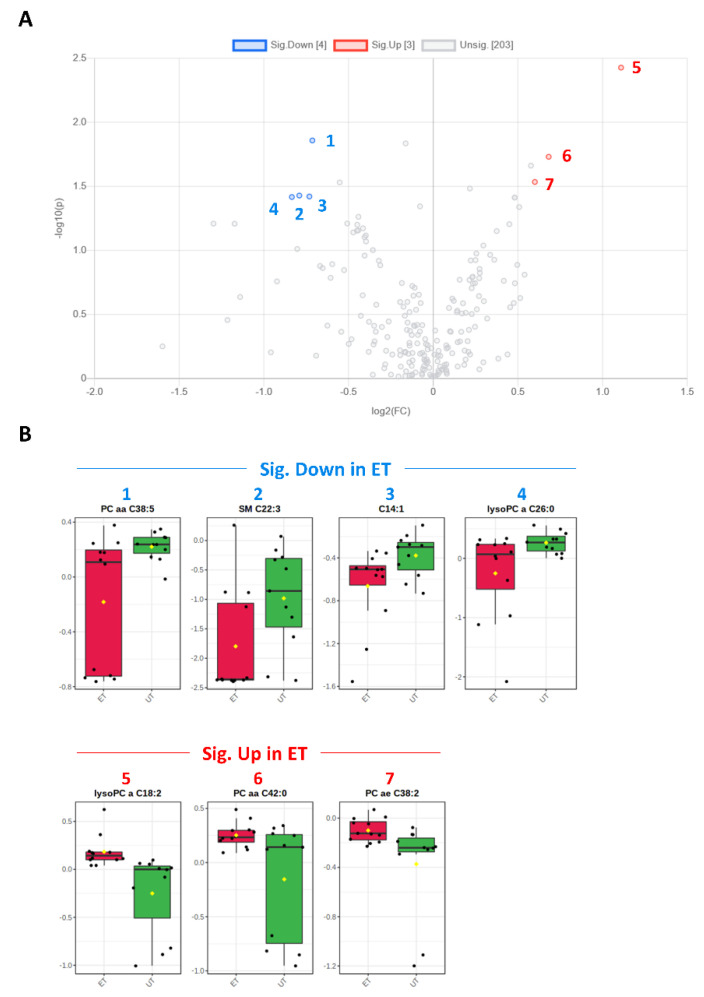
Volcano and box plots for significant hits as determined by the Biocrates Absolute IDQ p180 kit. (**A**) An unpaired fold change (FC) analysis was conducted to compare the absolute value of change between both groups. Therefore, the respective group means for each metabolite plasma concentration before data normalization have been used to calculate the fold change as well as the log_2_(FC) for the ratio of ET to UT, which is reflected by the x-axis. Simultaneous presentation of the value for −log_10_(*p*) value on the y-axis in the volcano plot enables easy detection of strongly and significantly deregulated candidates when *p* ≤ 0.05 during the unpaired, parametric *t*-test (e.g., −log_10_(*p*) ≥ 1.3). Metabolites that are significantly down-regulated in ET athletes (by a threshold of 1.5 set as minimum for the fold change) are numbered in blue color (1–4) and the up-regulated are in red (5–7), respectively. (**B**) Box plots allow visualization of plasma metabolite concentrations [µM] on the single-sample level (marked with black dots) with the 95% confidence interval around the group’s median indicated by the notch and by a yellow diamond highlighting the group’s mean.

**Figure 5 metabolites-12-00658-f005:**
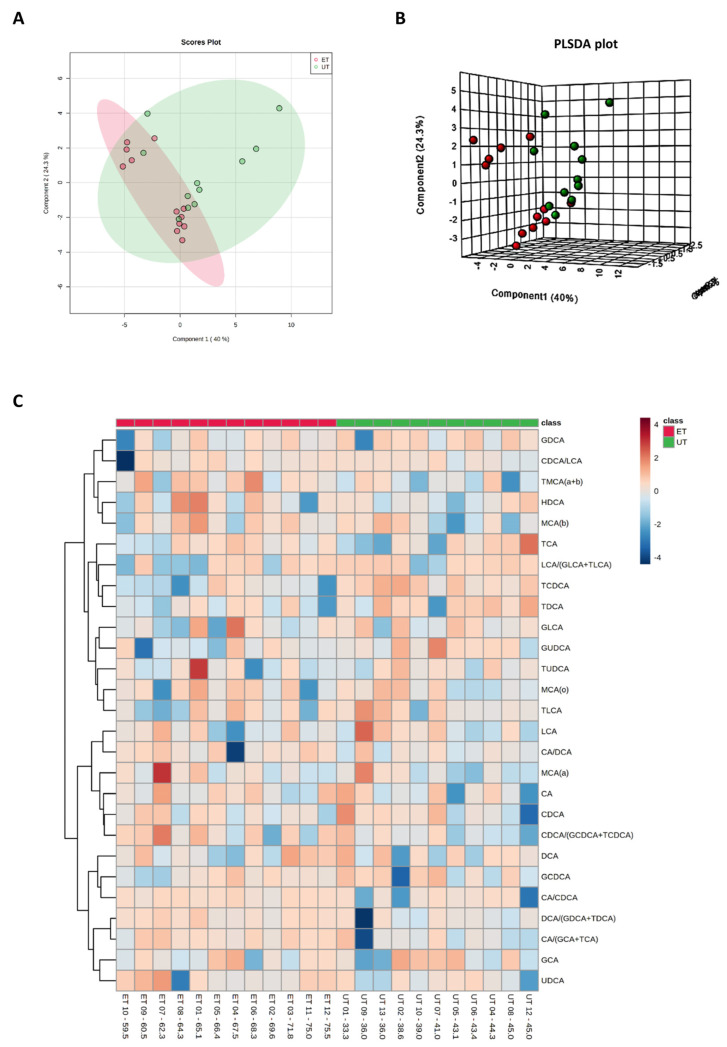
Component and heat map analyses of metabolites detected by the Biocrates Bile Acid Kit. (**A**) 2D principal component analysis (PCA) scores plot displaying individual samples from both groups, i.e., from ET (pink) and UT (green) subjects. (**B**) 3D multiple component partial least squares discriminant analysis (PLSDA) plot showing individual samples from both groups, i.e., from ET (red) and UT (green) subjects. (**C**) The heat map only shows the top 30 deregulated bile acids or ratios as listed on the vertical dimension. Bile acids (or their ratios) with high plasma levels are highlighted in orange to red, while the under-represented are colored in light to dark blue. Therefore, raw data, i.e., bile acid concentrations, had to be normalized by their median and log_2_-transformed prior horizontal arrangement according to their group affiliation, with increasing relative VO_2max_. Samples from ET athletes are located on the left side with red labels, and samples from UT subjects can be identified by their green labels.

**Figure 6 metabolites-12-00658-f006:**
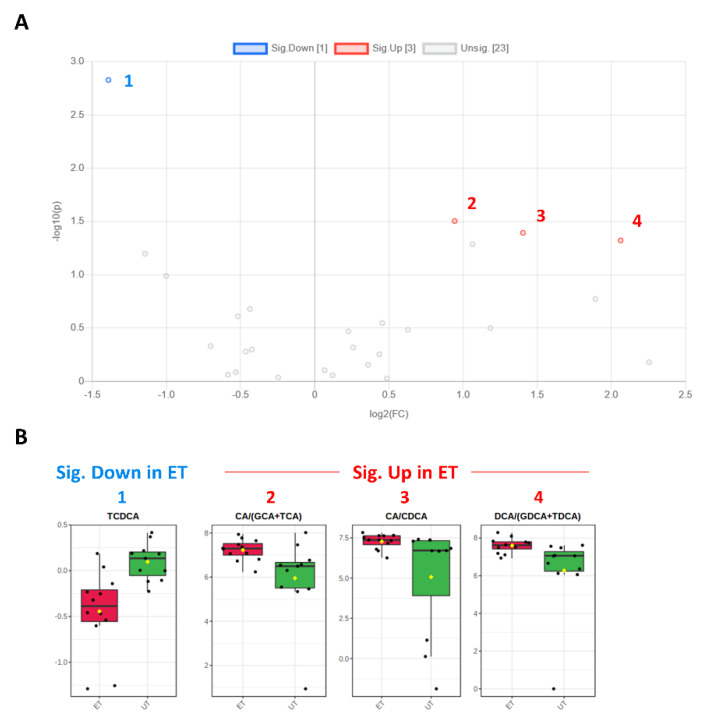
Volcano and box plots for significant hits as determined by the Biocrates Bile Acid Kit. (**A**) An unpaired fold change (FC) analysis was conducted to compare the absolute value of change between both groups. Therefore, the respective group means for each bile acid (or ratio) plasma concentration before data normalization have been used to calculate the fold change as well as the log_2_(FC) for the ratio of ET to UT, which is reflected by the x-axis. Simultaneous presentation of the value for −log_10_(*p*) value on the y-axis in the volcano plot enables easy detection of strongly and significantly deregulated candidates, when *p* ≤ 0.05 during the unpaired, parametric *t*-test (e.g., −log_10_(*p*) ≥ 1.3). Bile acids (or their ratios), which are significantly down-regulated in ET athletes (by a threshold of 1.5 set as minimum for the fold change), are numbered in blue color (1) and the up-regulated in red (2–4), respectively. (**B**) Box plots allow visualization of plasma bile acids concentrations [µM] (or their ratios) on the single-sample level (marked with black dots) with the 95% confidence interval around the group’s median indicated by the notch and by a yellow diamond highlighting the group’s mean.

**Figure 7 metabolites-12-00658-f007:**
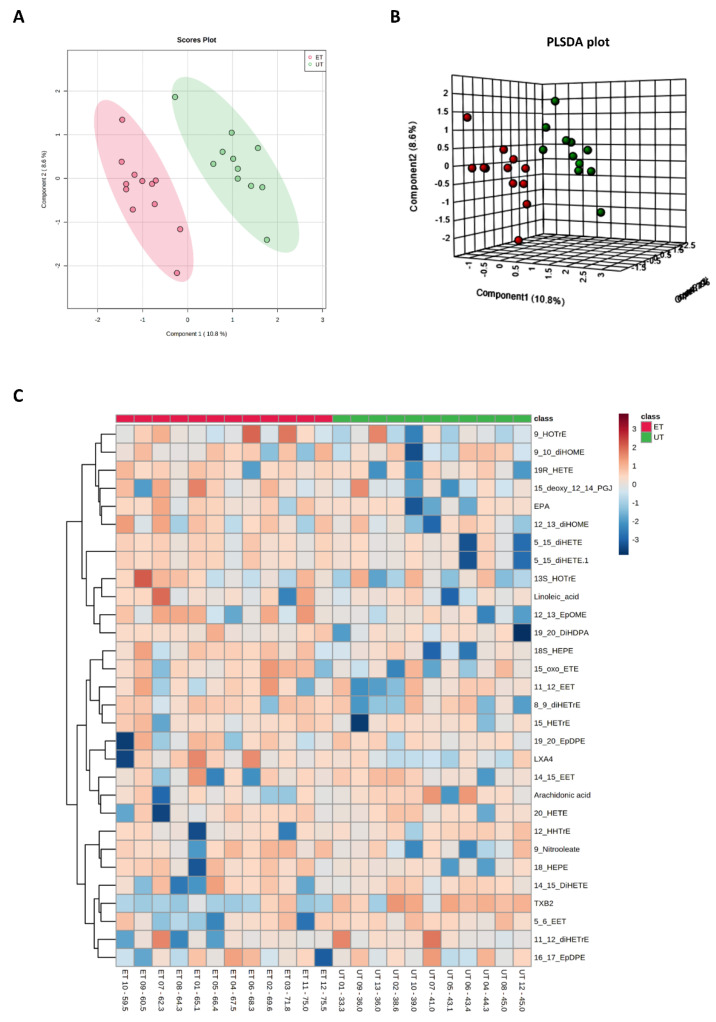
Component and heat map analyses of metabolites detected by the Biocrates Eicosanoid and Fatty acids Kit. (**A**) 2D principal component analysis (PCA) scores plot displaying individual samples from both groups, i.e., from ET (pink) and UT (green) subjects. (**B**) 3D multiple component partial least squares discriminant analysis (PLSDA) plot showing individual samples from both groups, i.e., from ET (red) and UT (green) subjects. (**C**) The heat map only shows the top 30 deregulated bile acids or ratios as listed on the vertical dimension. Bile acids (or their ratios) with high plasma levels are highlighted in orange to red, while the under-represented are colored in light to dark blue. Therefore, raw data, i.e., bile acid concentrations, had to be normalized by their median and log_2_-transformed prior horizontal arrangement according to their group affiliation, with increasing relative VO_2max_. Samples from ET athletes are located on the left side with red labels, and samples from UT subjects can be identified by their green labels.

**Figure 8 metabolites-12-00658-f008:**
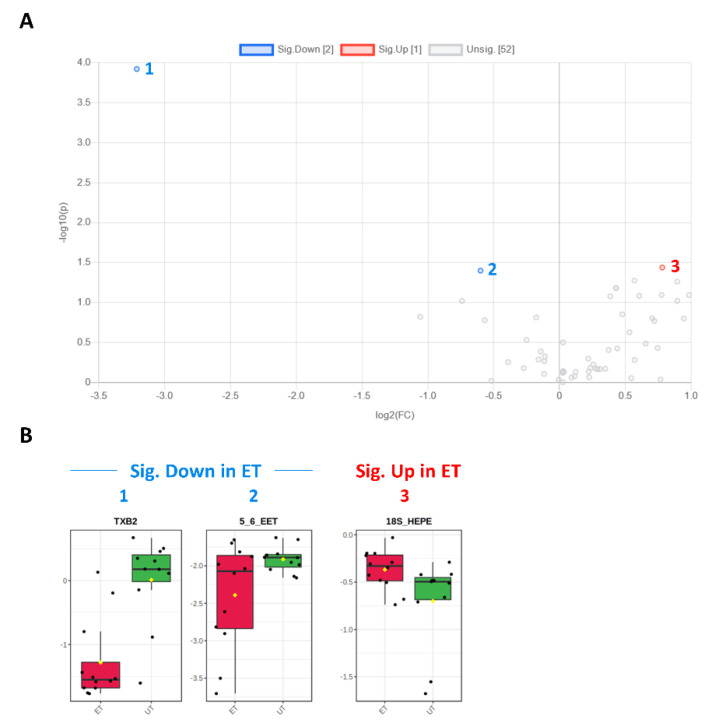
Volcano and box plots for significant hits as determined by the Biocrates Eicosanoid and Fatty acids Kit. (**A**) An unpaired fold change (FC) analysis was conducted to compare the absolute value of change between both groups. Therefore, the respective group means for each eicosanoid, fatty acid or PUFA plasma concentration before data normalization have been used to calculate the fold change as well as the log_2_(FC) for the ratio of ET to UT, which is reflected by the x-axis. Simultaneous presentation of the value for −log_10_(*p*) value on the y-axis in the volcano plot enables easy detection of strongly and significantly deregulated candidates when *p* ≤ 0.05 during the unpaired, parametric t-test (e.g., −log_10_(*p*) ≥ 1.3). Eicosanoids, fatty acids, or PUFAs which are significantly down-regulated in ET athletes (by a threshold of 1.5 set as minimum for the fold change) are numbered in blue color (1–2) and the up-regulated in red (3), respectively. (**B**) Box plots allow visualization of plasma eicosanoids, fatty acid, or PUFA concentrations (TXB2 [ng/mL]; 5,6-EET [area ratio]; 18S-HEPE [ng/mL]) on the single-sample level (marked with black dots), with the 95% confidence interval around the groups´ median indicated by the notch and by a yellow diamond highlighting the group’s mean.

**Figure 9 metabolites-12-00658-f009:**
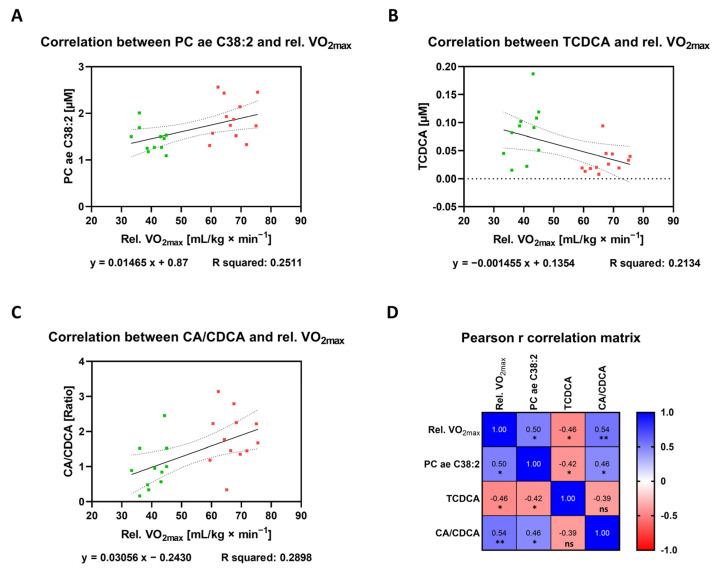
Correlation analyses to identify markers of cardio-pulmonary fitness. (**A**–**C**) Graphs display the relationship between relative VO_2max_ (x-axis) and the plasma concentration (y-axis) of distinct metabolites (or ratio) that fulfill two criteria: Firstly, they are deregulated between both study groups (Figure 4B, Figure 6B, and Figure 8B) and secondly, at the same time, they show a direct correlation with relative VO_2max_ regardless of their group affiliation (Table 1, Table 2 and Table 3). Correlation analyses were performed by a simple linear regression model computing the Pearson r correlation coefficient, R squared as well as the equation for the linear (filled line) slope, including the respective standard error of the mean (dotted line). Samples from the ET group are marked in red while UT subjects are colored in green. (**D**) The Pearson r correlation matrix represents the relationship between the above depicted metabolites (or ratio) amongst each other in addition to their previously stated correlation to relative VO_2max_. Statistical analyses of the two-tailed test assumes a Gaussian distribution revealing the color-coded Pearson r correlation coefficient as well as its significance level. ns: not significant; *: *p* ≤ 0.05; **: *p* ≤ 0.01.

**Table 1 metabolites-12-00658-t001:** Significant changes in sphingomyelins, acylcarnitines, and glycerophosphocholines in endurance-trained athletes compared to un-trained individuals.

Metabolite	HMDB ID	Category	FC	Log2 (FC)	p-Value	−Log10 (p-Value)	ET vs. UT
SM C22:3	0013468	SM	0.578	−0.791	0.037	1.428	Down (2)
C14:1	0002014	AC	0.602	−0.732	0.038	1.421	Down (3)
lysoPC a C18:2	0010386	GPL	2.157	1.109	0.004	2.428	Up (5)
PC aa C42:0	0008537	GPL	1.604	0.682	0.019	1.731	Up (6)
PC ae C38:2	0013436 0013431	GPL	1.516	0.600	0.029	1.535	Up (7)
PC aa C38:5	0008114	GPL	0.610	−0.714	0.014	1.859	Down (1)
lysoPC a C26:0	0029205	GPL	0.560	−0.835	0.038	1.417	Down (4)

**Table 2 metabolites-12-00658-t002:** Significant changes in bile acids (or their ratio) in endurance-trained athletes compared to un-trained individuals.

Metabolite (or Ratio)	HMDB ID	Category	FC	Log_2_ (FC)	*p*-Value	−Log_10_ (*p*-Value)	ET vs. UT
TCDCA	0000951	Bile acid	0.381	−1.392	0.002	2.828	Down (1)
CA/CDCA	0000619/ 0000518	Bile acid	2.645	1.403	0.041	1.393	Up (3)
CA/(GCA+TCA)	0000619/ (0000138 + 0000036)	Bile acid	1.923	0.943	0.031	1.504	Up (2)
DCA/(GDCA+TDCA)	0000626/ (0252868 + 0000896)	Bile acid	4.175	2.062	0.048	1.321	Up (4)

**Table 3 metabolites-12-00658-t003:** Significant changes in eicosanoids and polyunsaturated fatty acids in endurance-trained athletes compared to un-trained individuals.

Metabolite	HMDB ID	Category	FC	Log_2_ (FC)	*p*-Value	−Log_10_ (*p*-Value)	ET vs. UT
TXB2	0003252	Eicosanoid	0.107	−3.212	0.001	3.922	Down (1)
5,6-EET	0246887	Eicosanoid	0.660	−0.600	0.040	1.398	Down (2)
18S-HEPE	0257623 (S-Enantiomer)	PUFA	1.718	0.781	0.037	1.437	Up (3)

**Table 4 metabolites-12-00658-t004:** Correlation between different metabolites (or their ratios) and individual relative VO_2max_ including corresponding levels of significance, i.e. *p*-values of *: *p* ≤ 0.05; **: *p* ≤ 0.01.

Metabolite	HMDB ID	Category	Pearson R	95% CI	R Squared	*p* Value	Sig. Level
Asp	0000191	AA	−0.5046	−0.7589 to −0.1167	0.2546	0.0141	*
Glu	0000148	AA	−0.5268	−0.7715 to −0.1464	0.2776	0.0098	**
Cit/Orn	Cit: 0000904 Orn: 0000214	AA	0.5518	0.1808 to 0.7854	0.3045	0.0063	**
α-AAA	0000510	BA	−0.4722	−0.7403 to −0.07449	0.223	0.0229	*
Taurine	0000251	BA	−0.4512	−0.7280 to −0.04785	0.2035	0.0307	*
SM C18:0	0001348	SM	−0.421	−0.7100 to −0.01066	0.1773	0.0454	*
SM C18:1	0012101	SM	−0.5023	−0.7576 to −0.1136	0.2523	0.0146	*
C16	0000222	AC	−0.4414	−0.7222 to −0.03565	0.1948	0.035	*
PC aa C38:4	0008048	GPL	−0.4283	−0.7144 to −0.01958	0.1835	0.0414	*
PC ae C38:2	0013431 0013436	GPL	0.5011	0.1121 to 0.7570	0.2511	0.0149	*
PC ae C42:2	0013438	GPL	0.4517	0.0485 to 0.7283	0.204	0.0305	*
lysoPC a C18:1	0010385 0010408 0002815	GPL	0.4914	0.0993 to 0.7514	0.2415	0.0173	*
lysoPC_total_/PC_total_		GPL	0.5409	0.1657 to 0.7794	0.2926	0.0077	**
GCDCA	0000637	Bile acid	−0.4482	−0.7262 to −0.04413	0.2009	0.032	*
TCDCA	0000951	Bile acid	−0.462	−0.7343 to −0.06148	0.2134	0.0265	*
CA/CDCA	CA: 0000619 CDCA: 000518	Bile acid	0.5383	0.1620 to 0.7779	0.2898	0.0081	**
LXA4	0004385	PUFA	0.4313	0.0232 to 0.7161	0.186	0.0399	*
18-HEPE	0257623	PUFA	0.466	0.0552 to 0.7419	0.2171	0.0288	*

## Data Availability

All data are contained in the article and Appendix A. Original data can be obtained upon request.

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
