# Peer review of "Sustained Endurance Training Leads to Metabolomic Adaptation"

_metabolites, 2022, doi:10.3390/metabo12070658_

Round 1

Reviewer 1 Report

The authors investigated the metabolic profile of endurance athletes compared to untrained individuals based on selected metabolites from lipid and amino acid metabolism through a targeted metabolomics analysis. Authors used attractive data analysis such as principal component analysis, partial less square, discriminant analysis, etc. The figure which comes from this data analysis help readers to follow the text and make the manuscript very good-looking (Component and heat map analyses).

Although they used a relatively small number of individuals (with many monitored parameters), the authors were able to draw some conclusions from the large data set: results might represent an adaptation of lipid metabolism and account for the lowered cardiovascular risk profile of endurance athletes

Minors:

1. The authors could collect all the abbreviations and explain their meaning at the end of the manuscript.

2. The authors could give a scheme of metabolism: carbohydrates, lipids - fatty acids and amino acids in the discussion where they would indicate the modification of metabolism, i.e., possible points of metabolism regulation that are involved when metabolism goes to protecting athlete’s glycogen reserves - based on experimental data and their conclusions.

3. Bile acid concentrations are sensitive to the intestinal microbial flora, since intestinal microorganisms have not been analyzed, the authors could give a brief discussion of whether intestinal microorganisms possibly affect the bile acid concentration ratios.

Author Response

Reviewer #1

Comments and Suggestions for Authors

The authors investigated the metabolic profile of endurance athletes compared to untrained individuals based on selected metabolites from lipid and amino acid metabolism through a targeted metabolomics analysis. Authors used attractive data analysis such as principal component analysis, partial less square, discriminant analysis, etc. The figure which comes from this data analysis help readers to follow the text and make the manuscript very good-looking (Component and heat map analyses). Although they used a relatively small number of individuals (with many monitored parameters), the authors were able to draw some conclusions from the large data set: results might represent an adaptation of lipid metabolism and account for the lowered cardiovascular risk profile of endurance athletes

Minors:

  1. The authors could collect all the abbreviations and explain their meaning at the end of the manuscript.

We thank the reviewer for this suggestion and will provide a list of abbreviations at the end of the revised manuscript.

  1. The authors could give a scheme of metabolism: carbohydrates, lipids - fatty acids and amino acids in the discussion where they would indicate the modification of metabolism, i.e., possible points of metabolism regulation that are involved when metabolism goes to protecting athlete’s glycogen reserves - based on experimental data and their conclusions.

We appreciate this idea and edited the manuscript accordingly addressing this aspect. As we were only investigating concentrations of metabolites in the plasma but neither in organs nor individual cells, we can not draw conclusions on the overall change of all different metabolic processes in detail. For this purposes, we would have to include the analysis of carbohydrates, too. Nevertheless, the main finding of this manuscript is the clear difference between participants based on their exercise intensity and the subsequent metabolomic adaptation. Even with a selected set of detectable metabolites and a small n-number, sustained health-promoting, physiological alterations in endurance-trained athletes could be observed. Here, a metabolic shift of energy consumption involving fatty acid oxidation and lipid metabolism and a tight control of glycolytic pathways could be shown.

  1. Bile acid concentrations are sensitive to the intestinal microbial flora, since intestinal microorganisms have not been analyzed, the authors could give a brief discussion of whether intestinal microorganisms possibly affect the bile acid concentration ratios.

We thank the reviewer for this valuable comment. We will include a respective paragraph in the discussion but we would like to give a detailed answer as follows. Bile acids covered by the analysis are 10 primary bile acids which are produced by the liver (i.e. CA, GCA, TCA, CDCA, GCDCA, TCDCA, α-MCA, β-MCA, ω-MCA, TMCA(α+β)) as well as 10 secondary bile acids that are produced by gut bacteria from primary bile acids (i.e. DCA, GDCA, TDCA, LCA, GLCA, TLCA, UDCA, GUDCA, TUDCA, HDCA). Having a look into the data set on the bile acids we have to admit that the principal component analyses already show that the two groups of participants are not clearly separated from each other like it is the case for the other metabolites which we investigated, too. This already points out that we need an additional study which should include more participants but also have a stronger focus on bile acids with respect to the effects dominantly of exercise rather than nutrition. In this regard, it should be kept in mind, that the targeted metabolomics approach used in this manuscript can only give a first hint on the important network between the gut microbiome and physical exercise. Indeed, future studies are planned to address this topic as a central focus. Nevertheless, we can conclude from our data, that in three cases the concentration or ratios of primary bile acids is significantly different between ET and UT, while in one case we could observed a change in the ratio of secondary bile acids produced by gut-bacteria. In summary, we feel encourage by our data (and also the reviewers´ comments) to extend our scope of research to investigate the relationship between the microbiome, general health and physical exercise in the near future. This also includes an optimized experimental design as well as sophisticated technologies to particularly answer the question whether the intestinal microorganisms affect the concentration and ratios of certain bile acids under various physical conditions, i.e. untrained participants and endurance trained athletes. We have added a short section to the discussion.

Reviewer 2 Report

ET >= 59 mL/kgmin and UT <= 45 ml/kgmin. What about the people in between i.e. 45<X<59? Why specifically were those cut-offs where chosen?

Why is Section 3.1 in the results? It appears it should be in Methods.

Instead of looking at individual BAs, it is more useful to group them e.g. conjugated/unconjugated, taurine/glycine, 12-OH/non12-OH. For these

The only problem I have is with the logic. How do the authors conclude that the change in proteomics is due to exercise? The authors have chosen a selection of people where predominately the ET had a low BMI and the UT group had a high BMI.

The quality of graphics needs significant improvement. Especially when taking a screenshot with the autocorrect function on. 

Author Response

Reviewer #2

Comments and Suggestions for Authors

ET >= 59 mL/kgmin and UT <= 45 ml/kgmin. What about the people in between i.e. 45<X<59? Why specifically were those cut-offs where chosen?

We wanted to create a clear contrast between the group of trained endurance athletes and the group of untrained ones. In our experience, there are subjects in the middle performance range mentioned by the reviewer for whom it is not clearly definable whether they are endurance athletes or not. Therefore, we selected subjects below or above this limit. At the same time, we can already substantiate this approach with other studies.

Schild M, Ruhs A, Beiter T, Zügel M, Hudemann J, Reimer A, Krumholz-Wagner I, Wagner C, Keller J, Eder K, Krüger K, Krüger M, Braun T, Nieß A, Steinacker J, Mooren FC. Basal and exercise induced label-free quantitative protein profiling of m. vastus lateralis in trained and untrained individuals. J Proteomics. 2015 Jun 3;122:119-32. doi: 10.1016/j.jprot.2015.03.028.

Alack K, Krüger K, Weiss A, Schermuly R, Frech T, Eggert M, Mooren FC. Aerobic endurance training status affects lymphocyte apoptosis sensitivity by induction of molecular genetic adaptations. Brain Behav Immun. 2019 Jan;75:251-257. doi: 10.1016/j.bbi.2018.10.001.

Why is Section 3.1 in the results? It appears it should be in Methods.

We agree with the reviewer that the content in this section partially contains information which could also be provided in the method part. We previously published a paper with the same participants of the study, and we wanted to emphasize the importance of those findings also in direct relationship with the data presented now in the current manuscript. Therefore, we combined both, the information about the study design as wells as a summary of the already existing data set, in this first part directly at the beginning of the result section. From our point of view, the manuscript is thereby much more comprehensive for the reader.

Instead of looking at individual BAs, it is more useful to group them e.g. conjugated/unconjugated, taurine/glycine, 12-OH/non12-OH. For these

We appreciate this issue raised by the reviewer. We decided to perform the analyses based on the concentration of distinct bile acids. Ratios are only given when ratios could be identified that are supposed to give information about the breakdown of precursors into further products by metabolic conversion. Here, four cases have been presented in the respective heat map. As this point, we would also like to admit that the principal component analyses revealed a significant but very weak separation of both groups of participants. This might be explained by the fact that the study including the methodology of metabolite needs to be improved in terms of the number of participants as well as of the inclusion criteria that should also consider important aspects of bile acid metabolism, e.g. a special nutrition diet. Still our data can serve as a starting point to look into more detail in future studies. As one of the reviewer´s sentences seems to be incomplete “For these ….” , we hope that we answered the question as intended by the reviewer. In addition to this, the reviewer #1 gave a comment in the same direction and we encourage the reviewer #2 and the editor to look at our response to reviewer #1, too.

The only problem I have is with the logic. How do the authors conclude that the change in proteomics is due to exercise? The authors have chosen a selection of people where predominately the ET had a low BMI and the UT group had a high BMI.

We appreciate the reviewer´s concerns and would like to take the chance to address this topic in more detail. The two groups of participants differ in many physiological parameters but these all depend on the intensity of exercise which subsequently dictates all the other measures. The inclusion criteria are mostly driven by the VO2max value which serves (from our point of view) as the best marker of endurance performance. Because of this reason, all other inclusion criteria have been set very strict to allow a clear differentiation between the ET and UT participants. This also applies for the BMI in our study which is, as a matter of fact, directly affected by exercise and nutritional diet. We could rule out any underlying cause of a metabolic disease as far as the participants were aware of. We addressed the aspect of nutrition on the overall outcome as this is one of the most crucial potential confounder. So, if we assume that the BMI is completely dependent on the physical exercise and this is line which our data (i.e. we experienced a major difference between both groups in BMI), then we can also expect that the changes in the metabolic profile are also based on the exercise intensity and physical performance.

The quality of graphics needs significant improvement. Especially when taking a screenshot with the autocorrect function on.

We are agree with the reviewer and increased the resolution and the quality of the figures in the revised version of the manuscript.